# Topical Ozone as an Adjuvant Therapy in Wound Management: An Integrative Review

**DOI:** 10.3390/nursrep15120414

**Published:** 2025-11-25

**Authors:** Cristina Barroso Pinto, Adelino Pinto, Manuela Barroso, Telma Coelho, Sandra Costa

**Affiliations:** 1RISE-Health, Nursing School of Porto, University of Porto, 4200-072 Porto, Portugal; cristinabarroso@esenf.pt; 2Gaia/Espinho Local Health Unit, 4434-502 Vila Nova de Gaia, Portugal; adelino.m.c.pinto@gmail.com (A.P.); barrosomanuela@gmail.com (M.B.); 3Tãmega e Sousa Local Health Unit, 4564-007 Guilhufe, Portugal; telma_coelho25@hotmail.com; 4RISE-Health, School of Health, Santarém Plytechnic University, 2001-904 Santarém, Portugal

**Keywords:** wound, ozone, adjuvant therapies, nursing care, review

## Abstract

**Background/Objectives:** Wound management remains a clinical challenge, particularly in chronic and refractory conditions. Ozone, due to its antimicrobial, anti-inflammatory, and tissue-regenerative properties, has emerged as promising adjuvant therapy. This integrative re-view aimed to critically analyze the therapeutic effects, routes of administration, benefits, and limitations of ozone in wound treatment. **Methods:** The review followed the Joanna Briggs In-stitute methodology and the PRISMA 2020 guidelines. Studies were identified through compre-hensive search in the SCOPUS, CINAHL Ultimate, MEDLINE Ultimate, and MedicLatina data-bases, with no time restrictions. Inclusion criteria encompassed primary studies involving adults (≥18 years) with wounds treated with ozone. The methodological quality of the selected studies was assessed using the tools recommended by JBI. **Results:** Nine reports published between 2019 and 2025 met the inclusion criteria. The findings consistently demonstrated clinical benefits of ozone therapy, including accelerated wound healing, pain reduction, and infection control. The forms of application included ozonated water, ozonated olive oil, and gaseous ozone. However, heterogeneity was observed in ozone concentration, frequency, and treatment duration. The methodological quality of the included studies ranged from moderate to high. **Conclusions:** The available evidence indicates that ozone may represent promising adjuvant treatment for certain types of wounds; however, the quality and independence of the existing studies are limited, and the lack of standardized protocols as well as methodological variability restrict the generalizability of the findings. Therefore, more robust clinical trials are needed to strengthen the evidence base and support its clinical implementation.

## 1. Introduction

Wounds are defined as disruptions in the continuity of the skin, corresponding to tissue injuries typically caused by physical or mechanical damage. Such injuries may affect the epidermis, dermis, subcutaneous tissue, and deeper structures such as muscles, tendons, and bones, thereby compromising the integrity and function of the skin [1]. The etiology of these conditions is multifactorial, involving a combination of traumatic events, burns, pressure ulcers, surgical complications, and chronic diseases such as diabetes mellitus [2]. A comprehensive and meticulous wound assessment, encompassing parameters such as type, location, size, duration, level of contamination, and the condition of the surrounding skin, is essential for defining the therapeutic plan to be implemented and for continuously monitoring the effectiveness of the interventions performed [3,4].

The wound healing process is complex and dynamic, involving a sequential and organized series of biological events aimed at restoring tissue integrity. This process is traditionally divided into four phases: hemostasis, inflammation, proliferation, and maturation [3,5]. The effectiveness of healing depends on several intrinsic and extrinsic factors, including nutritional status, presence of comorbidities, tissue oxygenation, and the microbial load of the wound [3,5,6]. Chronic wounds, characterized by interruption or delay in the normal healing process, represent a significant challenge in clinical practice and are associated with high morbidity, reduced patient quality of life, and increased healthcare costs [7,8].

In recent years, the demand for innovative adjuvant therapies capable of accelerating healing and improving clinical outcomes has increased. Ozone therapy has emerged as a promising approach in this context. Ozone (O_3_) is an allotropic form of oxygen, composed of three oxygen atoms, recognized for its antimicrobial, anti-inflammatory, immunomodulatory, and antioxidant properties [9]. These characteristics make ozone particularly attractive for wound treatment, where infection prevention and modulation of the inflammatory response are crucial for promoting healing [10].

Ozone can be administered through different routes in clinical care practice, especially in the form of ozonated water or ozonated oil, which have been increasingly explored. In wound treatment, the topical application of ozone, in the form of ozonated water or oil, has been extensively studied. Recent studies suggest that the use of ozonated solutions can promote wound bed decontamination, stimulate angiogenesis, enhance tissue oxygenation, and accelerate the healing process [11,12,13,14]. Moreover, ozone therapy has shown potential in reducing wound-associated pain, contributing to improved patient comfort during treatment [15,16,17].

Despite these promising results, scientific evidence regarding the efficacy and safety of ozone therapy in wound treatment still presents limitations. Studies indicate significant benefits, such as enhanced healing and antimicrobial properties; however, they also underscore the need for further research with greater methodological rigor to validate these findings [18]. Additionally, the lack of standardization in application protocols, including ozone concentrations, frequency, and duration of treatment, complicates the comparison of results and limits the generalizability of conclusions [19].

In this context, conducting an integrative literature review becomes pertinent, aiming to synthesize the available evidence on the use of ozone in the treatment of wounds in adults. This review aims to critically analyze the therapeutic effects, routes of administration, benefits, and limitations of ozone in wound treatment.

## 2. Material and Methods

The present integrative review followed the methodology proposed by the Joanna Briggs Institute (JBI) [20], ensuring methodological rigor during study selection, critical appraisal, and synthesis. Additionally, the PRISMA 2020 Statement—Preferred Reporting Items for Systematic Reviews and Meta-Analyses [21] guidelines were followed, using the corresponding checklist to ensure transparency and completeness of the review. To guarantee the traceability and reproducibility of the research, this review was prospectively registered on the PROSPERO platform—International Prospective Register of Systematic Reviews, under registration number: CRD420251041214.

### 2.1. Research Question

The research question was structured using the PICO strategy (Population, Intervention, Comparison, Outcome), which allows for a structured and systematic approach in defining the key elements of the study. The target population (P) consists of adults with wounds; the intervention (I) is the application of topical ozone; the comparison (C) does not apply, as the study aims to evaluate the specific effects of ozone treatment; and the outcome (O) refers to the therapeutic effects, efficacy, and benefits associated with the use of ozone in wound treatment. Based on this framework, the guiding research question was: “What are the effects of ozone in the treatment of wounds in adults?”

This question supports a focused synthesis of the advantages, limitations, and clinical relevance of ozone therapy in adult wound care.

### 2.2. Studies Identification

The research was conducted between January and March 2025 and updated in June 2025, incorporating a broader set of databases to enhance the comprehensiveness and robustness of the review. The final search was performed across seven databases: SCOPUS, PubMed, Web of Science, LILACS, and EBSCOhost (including CINAHL Ultimate, MEDLINE Ultimate, and MedicLatina). Search strategies were carefully adapted to the indexing system and controlled vocabularies of each database (e.g., MeSH for PubMed/MEDLINE and DeCS for LILACS), ensuring high sensitivity and specificity.

The search was structured around three core concepts: (1) ozone therapy; (2) wound care; and (3) adjuvant or complementary treatments. Initial keywords such as “ozone”, “wounds”, and “adjuvant therapies” were expanded using both controlled descriptors (e.g., “Ozone”[MeSH], “Wound Healing”[MeSH], “Complementary Therapies”[MeSH]) and natural language synonyms (e.g., “ozonated oil”, “topical ozone”, “diabetic foot ulcer”).

Boolean operators “AND” and “OR” were used to build complex and tailored search phrases for each database. Example: (“Ozone” OR “ozonated water” OR “ozone gas”) AND (“wound healing” OR “chronic wounds” OR “diabetic ulcer”) AND (“topical” OR “adjuvant therapy” OR “complementary therapy”).

Language filters were applied (Portuguese, English, and Spanish), but no restrictions were imposed regarding publication date, to capture the widest possible range of relevant literature. The refinement of the search strategy was guided by peer-review feedback and aligned with PRISMA 2020 recommendations for systematic reviews [21]. No publication date restrictions were imposed.

Table 1 summarizes the search strategy for each database used, including the search terms.

### 2.3. Inclusion and Exclusion Criteria

Inclusion criteria comprised primary studies with no time limit, full-text availability, involving adults (≥18 years) with acute or chronic wounds, evaluating topical ozone applications (ozonated water or oil), and reporting clinical outcomes.

Excluded studies included those involving systemic or inhalational ozone, pediatric or obstetric populations, lacking methodological clarity, as well as reviews, duplicates, and non-scientific publications.

### 2.4. Study Selection

Two reviewers independently selected the studies using EndNote^®^ (version 20) to remove duplicates and Rayyan^®^ (free web-based version) (https://www.rayyan.ai, accessed on 1 July 2025) for blinded screening. Titles and abstracts were screened according to the predefined eligibility criteria. In cases of disagreement between reviewers, a third reviewer was consulted to reach consensus.

Full texts of potentially eligible articles were retrieved and evaluated in depth. A standardized data extraction form was used to collect information on the author, year, country, objective, study design, sample size, intervention characteristics, and main results.

### 2.5. Quality Appraisal

The methodological quality of the studies included was assessed using the standardized critical appraisal tools developed by the Joanna Briggs Institute (JBI), tailored to each study design (e.g., RCTs, quasi-experimental, observational, case reports). The appraisal was performed independently by two reviewers, and discrepancies were resolved through discussion or consultation with a third reviewer. The appraisal focused on key elements such as bias control, clarity of intervention, outcome measures, and ethical conduct.

### 2.6. Data Analyses

Data synthesis was performed using inductive thematic analysis, involving an in-depth reading of the articles and coding of relevant content, which allowed the identification of emergent categories. The presentation of the results is organized according to the thematic categories identified, providing a comprehensive understanding of the effects and benefits attributed to ozone wound treatment.

As this is a literature review, submission to a Health Ethics Committee was not required. Nevertheless, scientific rigor and academic integrity were maintained throughout the process, with strict adherence to copyright standards and proper referencing of all consulted sources.

## 3. Results

The PRISMA 2020 flow diagram illustrates the selection process (Figure 1).

As shown in Figure 1, studies were identified through searches in four electronic databases: SCOPUS (n = 2), CINAHL Ultimate (n = 15), MEDLINE Ultimate (n = 17), MedicLatina (n = 2), PubMed (n = 38), Web of Science (n = 24), and LILACS (n = 11), yielding a total of 106 records. After removing 28 duplicate entries, 78 records remained for screening. In the initial phase, the titles were reviewed, and 48 records were excluded for not meeting the predefined inclusion criteria. The remaining 30 abstracts were assessed, resulting in the exclusion of an additional 16 studies.

Fourteen full-text articles were then evaluated in depth, and five were excluded because the intervention did not meet the eligibility criteria. During the study selection process, it became evident that some publications originated from the same research program, which was subdivided into specific areas. To clarify the potential overlap of participants, the original investigators were contacted; however, it was not possible to unequivocally confirm the extent of duplication. In light of this limitation, the findings were considered as nine reports derived from five distinct studies, acknowledging the possibility of partial participant overlap.

### 3.1. Quality Appraisal and Assessment of Risk of Bias

The methodological quality of the included studies was critically appraised using the JBI (Joanna Briggs Institute) tools appropriate for each study design (randomized controlled trials, quasi-experimental studies, observational studies, or qualitative research). This appraisal facilitated the identification of potential risks of bias and contributed to the rigor and validity of the evidence synthesis.

The results of this assessment are presented in Table 2, which outlines each study’s design and the JBI tool used.

Table 3 summarizes the results of the critical appraisal conducted using the JBI tools appropriate for each study design.

Each study was scored based on the specific criteria of the JBI tool, and a methodological quality rating was subsequently calculated. The assessment results revealed scores ranging from 60% to 90%, representing moderate to high methodological quality. Although some studies presented limitations in the description of randomization processes, bias control, or methodological design—particularly in case studies or single-case reports, they were nonetheless retained in this review.

This decision was based on the premise that integrative reviews allow for the inclusion of different types of studies (quantitative, qualitative, and case studies), provided they undergo a rigorous critical appraisal process [28,29]. Moreover, in emerging fields such as the use of ozone in wound treatment, there is often a lack of robust clinical trials, making it pertinent to consider studies with diverse designs to gain a broader understanding of the topic [30,31].

### 3.2. Data Extraction

To ensure a clear and systematic presentation of the data extracted from the studies included in this integrative review, two complementary summary tables were prepared. Table 4 outlines the general characteristics of the included studies, such as the author(s), year of publication, country of origin, and the objective of each study. Table 5 provides a structured synthesis of the methodological nature of the studies, the types of wounds treated, the specific characteristics of the ozone intervention, and the main outcomes reported.

The studies included in this review were conducted in multiple countries and covered a range of clinical conditions and ozone treatment approaches. Their general characteristics are detailed in Table 4.

To better understand the strengths and limitations of the current evidence base, it is essential to examine the methodological designs and objectives of the studies included in this review. Specifically, two studies followed a randomized design [22,23], one employed a quasi-experimental approach [26], and two were analytical observational studies [17,24], the latter being a recent retrospective cohort focused on diabetic foot ulcers. Additionally, three case reports [15,26,27] and one small-sample experimental study [25] were included, reflecting the emerging and exploratory nature of research in this field.

The objectives of the included studies varied, although all focused on evaluating the efficacy of ozone therapy in the treatment of different types of wounds. These ranged from chronic ulcers and diabetic foot syndrome to oral leukoplakia and bullous pemphigoid, indicating the broad spectrum of potential clinical applications for ozone-based interventions.

The included studies comprised randomized controlled trials, quasi-experimental studies, observational studies, and case reports, reflecting the exploratory nature of research in this emerging field. The clinical conditions addressed were diverse, encompassing diabetic foot ulcers, venous ulcers, oral leukoplakia, post-surgical infections, bullous pemphigoid, and gout-related wounds.

Regarding the intervention, different forms of ozone application were used—ozonated water, ozonated olive oil, and gaseous ozone—with varied concentrations, frequencies, and durations. For example, study S2 [23] demonstrated enhanced healing with 60 µg/mL ozone compared to 30 µg/mL, and study S4 [24] reported significant clinical and biochemical improvements after 12 sessions of gaseous ozone over 12 weeks, including reduced inflammatory markers and increased angiogenic activity.

Despite this heterogeneity, most studies reported positive clinical outcomes, including accelerated wound healing, pain reduction, infection control, and enhanced tissue regeneration. These preliminary results reinforce the therapeutic potential of ozone in wound care, particularly as an adjuvant approach. However, the marked variability in ozone concentration, application methods, and study design still limits comparability across findings and underscores the urgent need for standardized protocols and high-quality clinical trials, ideally supported by international guidelines.

## 4. Discussion

Based on the evidence extracted from the selected studies, four thematic categories were identified: (1) Therapeutic effects and clinical outcomes, (2) Routes of administration and characteristics of intervention, (3) Advantages and clinical applicability, and (4) Limitations and methodological variability.

### 4.1. Therapeutic Effects and Clinical Outcomes

The therapeutic effects of ozone were consistently positive across the reviewed studies, with observed improvements in pain reduction, infection control, wound contraction, modulation of inflammation, and overall healing progression. A higher healing rate and significant pain reduction were demonstrated in patients with bullous pemphigoid treated with ozonated water [22]. Similarly, reductions in ulcer size and pain were reported in patients with diabetic foot, venous ulcers, and chronic ulcers, highlighting the clinical effectiveness of local ozone applications [15,16,17,23].

Notably, one of the included studies provided strong evidence not only of clinical improvement in diabetic foot ulcers but also of biochemical modulation. Patients treated with short-term gaseous ozone showed reduced inflammatory markers (CRP, IL-6, TNF-α) and increased angiogenic and antioxidant factors (VEGF and SOD), reinforcing the multi-mechanistic benefits of ozone therapy [24].

Some studies also emphasized the contribution of ozonated oil to epithelialization and infection control [25,26]. Although some of the studies involved isolated cases, the consistency in reported clinical benefits strengthens the evidence supporting the therapeutic potential of ozone. This supports the hypothesis that ozone therapy may act by modulating oxidative stress, stimulating local immune responses, regulating inflammation, and promoting angiogenesis—mechanisms that are particularly relevant in chronic or ischemic wounds. Altogether, these findings suggest that ozone therapy may play a critical role in improving wound healing outcomes, especially in complex, hard-to-heal clinical scenarios.

### 4.2. Routes of Administration and Characteristics of Intervention

The reviewed studies presented heterogeneous approaches to ozone application, including ozonated water, ozonated olive oil (with or without vitamin E), and medical ozone gas. The frequency and duration of treatments ranged from daily applications over 14 days [22] to up to 30 sessions across several weeks [16,23], reflecting considerable variability in clinical protocols. The concentration of ozone, when reported, varied from 30 µg/mL to 60 µg/mL [23], with better outcomes observed at higher concentrations. However, no consistent dose-response pattern could be confirmed across studies.

One study involving diabetic foot ulcers employed a short-term protocol with medical ozone gas and demonstrated both clinical healing and significant systemic effects, even without prolonged or high-frequency exposure [24]. This highlights that treatment efficacy may depend not only on dose and duration, but also on patient profile and wound characteristics.

Despite these differences, the commonality in local administration routes suggests a preference for direct application to wound sites, facilitating targeted action, improved oxygenation, and antimicrobial effects. These findings are consistent with recent literature emphasizing the oxidative, anti-inflammatory, and immunomodulatory properties of topical ozone in tissue regeneration [32,33,34]. Additionally, the ability of ozone to enhance local tissue oxygenation and stimulate angiogenesis may contribute significantly to wound healing, particularly in ischemic or chronic wounds.

### 4.3. Advantages and Clinical Applicability

The advantages of ozone therapy include its non-invasiveness, cost-effectiveness, and potential to enhance wound healing in complex clinical scenarios. Evidence from multiple studies [15,16,17,23] highlights its effectiveness in managing chronic or refractory wounds, supporting its role as an adjunctive modality in standard wound care. Furthermore, its combination with other therapeutic approaches—such as cryosurgery [25] or vitamin E supplementation [26]—demonstrates a high degree of adaptability to diverse clinical settings. This versatility reinforces the value of ozone therapy as a complementary strategy within multidisciplinary wound care protocols.

These findings corroborate previous reviews suggesting that ozone therapy may reduce antibiotic use, treatment time, and hospitalization length, particularly in infections and chronic wound management [32,34,35,36]. In addition, clinical guidelines emphasize the importance of protocol standardization, safety assurance, and consistent efficacy evaluation, reinforcing its role in integrative and evidence-informed wound care frameworks [37]. Nonetheless, these potential benefits should be interpreted with caution given the limited number of high-quality randomized trials and the predominance of exploratory or observational designs. As such, ozone therapy could play a significant role in addressing the growing concerns of antibiotic resistance and the need for innovative, adjunctive wound healing strategies.

### 4.4. Study Limitations and Methodological Variability

Despite the promising results, the studies reviewed exhibit several methodological limitations that must be acknowledged. These include small sample sizes, heterogeneous study designs, limited use of control groups in non-randomized studies, and insufficient or inconsistent descriptions of ozone preparation and application protocols. The methodological quality scores, ranging from 60% to 90%, indicate moderate to high quality overall, but also reveal weaknesses in randomization, blinding, and control of confounding variables, particularly in quasi-experimental and observational studies.

Furthermore, the lack of standardization regarding treatment duration, ozone concentrations, delivery method, and outcome assessment tools hinders comparability across studies and limits the generalizability of the findings. Although one study [24] demonstrated methodological rigor within a retrospective design—reporting biomarker data and clinical endpoints in a controlled cohort—it still lacked randomization and blinding, underscoring the overall fragility of the evidence base. This heterogeneity reflects the preliminary nature of research in this field and highlights the urgent need for scientific consensus and harmonized clinical protocols. These gaps underscore the importance of conducting more robust, multicenter clinical trials employing standardized methodologies, clearly defined endpoints, and long-term follow-up. Future studies should not only address efficacy, but also report safety outcomes, cost-effectiveness analyses, and patient-reported measures, aiming to inform clinical guidelines and support broader implementation of ozone therapy in wound management.

Some included studies originate from the same research groups and share the same ethical approval reference numbers. As this is an integrative review, we relied on the methodological appraisal of the published studies and did not independently verify their ethical documentation. This represents a limitation that should be considered when interpreting the findings.

Additionally, although we contacted the investigators responsible for the original studies, it was not possible to clearly determine the extent of participant overlap across some of the publications. For this reason, this review reports the evidence as nine reports stemming from five studies, and this limitation should be considered when interpreting the findings.

## 5. Conclusions

The findings of this integrative review indicate that ozone, in its various forms of application (ozonated water, ozonated oil, and gaseous ozone), demonstrates potentially beneficial therapeutic effects in wound treatment. Its antimicrobial, anti-inflammatory, antioxidant, and tissue-regenerative properties are particularly noteworthy. The evidence reviewed consistently shows preliminary clinical and, in some cases, biochemical benefits, including accelerated wound healing, pain reduction, infection control, and modulation of inflammatory biomarkers—all of which contribute to improved clinical outcomes and enhanced patient quality of life.

These results support the exploratory integration of ozone therapy into clinical practice, particularly as an adjunctive strategy within multidisciplinary wound care frameworks. However, the methodological heterogeneity observed across the studies—particularly regarding ozone concentration, treatment duration, delivery route, and outcome measures—underscores the need for more rigorous, multicenter, and standardized clinical trials. Such trials are essential to further validate the therapeutic potential of ozone and to establish evidence-based guidelines for its effective, safe, and scalable use in clinical settings.

## Figures and Tables

**Figure 1 nursrep-15-00414-f001:**
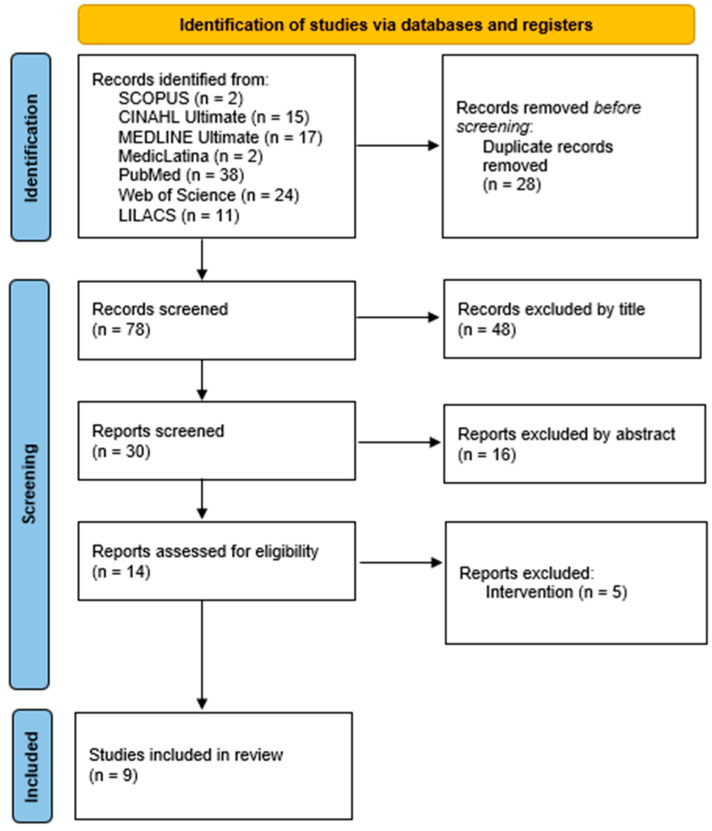
PRISMA 2020 flow diagram.

**Table 1 nursrep-15-00414-t001:** Search strategies and results by database.

Database	Search Strategy	Results Retrieved
SCOPUS	((adult) AND (“ozone”) AND (“wounds”) AND (“therapeutic effects” OR “therapeutic efficacy”))	2
CINAHL Ultimate	((adults OR aged OR elderly) AND (“ozone therapy”) AND (“therapeutic effect” OR “therapeutic efficacy”) AND (“wounds” OR “chronic wounds”))	15
MEDLINE Ultimate	(“Ozone”[MeSH] OR “ozone therapy”) AND (“Wound Healing”[MeSH] OR “Skin Ulcer”[MeSH]) AND (“Topical Administration”[MeSH] OR “topical ozone”)	17
MedicLatina	(“ozonoterapia” OR “ozono tópico”) AND (“feridas” OR “úlceras”) AND (“terapia adjuvante” OR “tratamento complementar”)	2
PubMed	(“Ozone”[MeSH Terms] OR “ozonated oil” OR “ozonated water” OR “ozone gas”) AND (“Wound Healing”[MeSH Terms] OR “chronic wounds” OR “diabetic foot ulcer”) AND (“Topical Administration”[MeSH] OR “adjuvant therapy” OR “complementary therapy”)	38
Web of Science	TS=(“ozone” AND “wound healing” AND (“topical” OR “adjuvant” OR “complementary therapy”))	24
LILACS	(“ozônio” OR “ozonioterapia”) AND (“feridas” OR “úlceras crônicas”) AND (“terapia adjuvante” OR “cuidados complementares”) [DeCS terms applied]	11

**Table 2 nursrep-15-00414-t002:** Study design and JBI tool used.

Paper	Study Design	JBI Tool
Li, L. et al. (2025) [22]—Evaluating the therapeutic efficacy of ozone liquid dressing in healing wounds associated with bullous pemphigoid	Experimental study	JBI Critical Appraisal Checklist for Randomized Controlled Trials
Pasek, J. et al. (2024) [23]—Impact of ozone concentration on the treatment effectiveness of diabetic foot syndrome: a pilot single-centre study	Quasi-Experimental Study	JBI Critical Appraisal Checklist for Quasi-Experimental Studies
Pasek, J. et al. (2024) [17]—Effect of Treatment of Neuropathic and Ischemic Diabetic Foot Ulcers with the Use of Local Ozone Therapy Procedures	Observational study	JBI Critical Appraisal Checklist for Analytical Observational Studies
Sun et al. (2024) [24]—Evaluation of the healing potential of short-term ozone therapy for the treatment of diabetic foot ulcers	Retrospective cohort study	JBI Critical Appraisal Checklist for Analytical Cohort Studies
Pasek, J. et al. (2023) [16]—Topical Hyperbaric Oxygen Therapy Versus Local Ozone Therapy in Healing of Venous Leg Ulcers	Randomized Controlled Trials	JBI Critical Appraisal Checklist for Randomized Controlled Trials
Darenskaya, M.A. et al. (2023) [25]—Effectiveness of Combined Application of Cryosurgical Method and Local Ozone Therapy in Patients with Oral Leukoplakia	Experimental study	JBI Critical Appraisal Checklist for Experimental Studies
Marinova, P.G. (2023) [26]—Ruptured Ulcerated and Inflamed Gout Tophi with Deep Soft Tissue Infection of Left Foot—Case Treated with Local Ozone Therapy	Case report	JBI Critical Appraisal Checklist for Case Reports
Pasek, J. et al. (2022) [15]—Ozone Therapy in the Comprehensive Treatment of Leg Ulcers: Case Report	Case report	JBI Critical Appraisal Checklist for Case Reports
Buric et al. (2019) [27]—Severe spinal surgery infection and local ozone therapy as complementary treatment: A case report	Case report	JBI Critical Appraisal Checklist for Case Reports

**Table 3 nursrep-15-00414-t003:** Critical appraisal of methodological quality.

Paper	Score Obtained	Maximum Score	Percentage (%)	Justification
Li et al. (2025) [22]	8	10	80%	Well-structured study, but with some shortcomings in detailing the randomization and allocation criteria.
Pasek et al. (2024) [23]	7	10	70%	Minor flaws in the description of how participants were allocated to different ozone concentrations.
Pasek et al. (2024) [17]	6	10	60%	Some issues regarding control of selection and confounding bias.
Sun et al. (2024) [24]	8	10	80%	Retrospective study with reasonable control group and appropriate statistical analysis; limitations in randomization and blinding
Pasek et al. (2023) [16]	9	10	90%	Well-conducted study, although with limitations in bias control.
Darenskaya et al. (2023) [25]	8	10	80%	Good outcomes, but some limitations in randomization and bias control.
Marinova (2023) [26]	6	10	60%	Interesting application but limited due to single-case study design.
Pasek et al. (2022) [15]	7	10	70%	Limited conclusions due to case study format.
Buric et al. (2019) [27]	7	10	70%	Positive clinical outcomes, but limited generalizability due to being a single case report

**Table 4 nursrep-15-00414-t004:** General characteristics of the selected studies.

Study	Author	Year	Country	Aim of the Study
S1	Li et al. [22]	2025	China	To evaluate the therapeutic efficacy of liquid ozone dressings in patients with bullous pemphigoid
S2	Pasek et al. [23]	2024	Poland	To assess the impact of ozone concentration on the effectiveness of diabetic foot syndrome treatment
S3	Pasek et al. [17]	2024	Poland	To evaluate the healing of neuropathic and ischemic diabetic ulcers with local ozone
S4	Sun et al. [24]	2024	China	To evaluate the clinical outcomes and inflammatory biomarkers in diabetic foot ulcers treated with short-term topical ozone therapy
S5	Pasek et al. [16]	2023	Poland	To compare the therapeutic efficacy of topical hyperbaric oxygen therapy versus ozone therapy in venous ulcers
S6	Darenskaya et al. [25]	2023	Russia	To evaluate the efficacy of cryosurgery combined with ozone in oral leukoplakia
S7	Marinova [26]	2023	Bulgaria	To report the treatment of an ulcerated and infected gouty tophus with ozone
S8	Pasek et al. [15]	2022	Poland	To report a case of chronic ulcer treated with ozone
S9	Buric et al. [27]	2019	Italy	To report a post-surgical infection treated with ozone as adjunctive therapy

**Table 5 nursrep-15-00414-t005:** Synthesis of the selected studies.

Study	Methodology and Sample Size	Type of Wound	Ozone Intervention	Main Results
S1	Experimental study with control group (n = 120)	Bullous pemphigoid	Ozonated water; daily application; up to 14 days	Higher healing rate; reduced pain and infection
S2	Quasi-Experimental Study (n = 50)	Diabetic foot	O3 at 30 µg/mL (group 1) and 60 µg/mL (group 2); 30 sessions of 30 min	Greater wound area and pain reduction in group 2
S3	Observational study (n = 90)	Neuropathic and ischemic diabetic ulcers	Local O3; number of sessions not specified	Significant wound reduction; higher efficacy in neuropathic ulcers
S4	Retrospective cohort study (n = 89)	Diabetic foot ulcers	Short-term topical ozone therapy (gaseous ozone); evaluated over 12 weeks	Improved healing rate, reduced inflammation (↓ CRP, IL-6, TNF-α), ↑ VEGF and SOD; fewer reinfections and reoperations
S5	Randomized Controlled Trials (n = 114)	Venous ulcer	30 sessions of local ozone	Reduction of ulcer area and pain; less effective than oxygen therapy
S6	Experimental study (n = 33)	Verrucous oral leukoplakia	Ozonated olive oil after cryosurgery; 14 days	Pain and edema reduction; increased epithelialization
S7	Case report	Ulcerated and infected gouty tophus	Ozonated olive oil with vitamin E	Progressive improvement in infection and healing
S8	Case report	Chronic ulcer post-orthopedic surgery	30 sessions of local ozone	Complete healing; pain relief and improved quality of life
S9	Case report	Spinal surgical infection	Ozone gas injected twice/week for 3 weeks	Total wound healing; no recurrence after 1 year

The symbols: ↓ indicates a decrease and ↑ indicates an increase.

## Data Availability

No new data were created or analyzed in this study.

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
