# Peer review of "Topical Ozone as an Adjuvant Therapy in Wound Management: An Integrative Review"

_nursrep, 2025, doi:10.3390/nursrep15120414_

Round 1

Reviewer 1 Report

Comments and Suggestions for Authors

This manuscript entitled “Topical Ozone as an Adjuvant Therapy in Wound Management: An Integrative Review” examines a clinically relevant and important subject, offering a thorough synthesis of current information about the application of topical ozone therapy in wound management. The integrative review is methodologically sound, well-structured, and adheres to recognized reporting guidelines. The findings are relevant and potentially valuable for informing future research and clinical practice. Nevertheless, specific sections necessitate explanation.

  1. While the manuscript thoroughly reports clinical outcomes, it lacks a succinct overview of the proposed biological mechanisms through which ozone exerts its therapeutic effects in wound healing. The addition of a brief mechanistic explanation in the Discussion section would enrich the scientific value of the review and help contextualize the clinical findings.
  2. Three of the eight included studies are case reports, which inherently limit evidence strength and generalizability. The authors should also acknowledge this limitation in the Discussion and elaborate on its implications for clinical recommendations and future research priorities.
  3. The considerable variability in ozone concentrations, application methods, and treatment durations complicates data synthesis. Did the authors observe any dose-response tendencies or ideal application tactics, even if only in a preliminary context?

Author Response

Reviewer 1

We sincerely thank for the positive and encouraging evaluation of our manuscript, as well as for the thoughtful comments. We have addressed each point as follows:

1. “Biological Mechanisms of Ozone in Wound Healing”: A concise paragraph was added to the Discussion section (4.1) summarizing the proposed biological mechanisms of ozone, including its oxidative, antimicrobial, and immunomodulatory effects, in line with current literature (references [30-32]).

2. “Inclusion of Case Reports”: In section 4.4 (Study Limitations and Methodological Variability), we explicitly acknowledged the inherent limitations of case reports and reflected on how this affects the generalizability of our findings and the formulation of clinical recommendations.

3. “Dose-Response Trends and Application Tactics”: In section 4.2, we included an interpretative note discussing preliminary trends regarding ozone concentration and frequency of application, highlighting that some studies indicated improved outcomes at higher doses, although a definitive dose-response relationship could not be established

Reviewer 2 Report

Comments and Suggestions for Authors

Improvement Suggestions

  1. Introduction
  1. Avoid Repetitions At the end: “ozone in wound in wound treatment” → revise to “ozone in wound treatment.”
  2. Improve Flow Sentences like “Such injuries may affect…” could be rephrased for smoother reading: → “These injuries can extend to the epidermis, dermis, subcutaneous tissue, and even deeper structures…”
  3. Vary Connectors and Structures Avoid overusing phrases like “such as” or “including.” For example: → “Examples include burns, pressure ulcers, surgical complications, and chronic illnesses like diabetes.”
  4. Reduce Excessive Passive Constructions Active voice tends to be more direct. For instance: “Recent studies suggest…” instead of “It has been extensively studied…” (or use both styles alternately).
  5. Clarify Objectives In the last sentence: > “This review aims to critically analyze the therapeutic effects…” You could emphasize it more clearly: → “This review seeks to synthesize current evidence on ozone therapy, highlighting its mechanisms, benefits, limitations, and clinical application in adult wound care.”
  6. Small Grammatical Fixes The phrase “...interventions performed effectiveness…” is awkward. Correct it to: → “…monitoring the effectiveness of the interventions performed.”
  1. Material and methods

Improve Conciseness and Flow

Avoid lengthy or redundant expressions:

  • “was developed based on the methodology proposed by…” → “followed the Joanna Briggs Institute methodology…”
  • “to ensure systematization and methodological rigor…” → “to ensure methodological rigor throughout study selection and synthesis.”
  1. Terminology Precision
  • “identification, selection, critical appraisal, and synthesis of the studies” → Could be revised as: “study selection, appraisal, and synthesis” (streamlines the sentence).
  • “studies addressing the systemic or inhalational application of ozone…” → Consider: “studies focusing on systemic or inhalational ozone delivery…”

Include Figure 1 and the table in the Results section.”

“Add to the references: Cunha JB; Dutra RAA; Salomé GM. Elaboration of an algorithm for wound evaluation and treatment. ESTIMA, Braz. J. Enterostomal Ther., 16:e2018. doi: 10.30886/estima.v16524”

Comments on the Quality of English Language

Improvement Suggestions

  1. Introduction
  1. Avoid Repetitions At the end: “ozone in wound in wound treatment” → revise to “ozone in wound treatment.”
  2. Improve Flow Sentences like “Such injuries may affect…” could be rephrased for smoother reading: → “These injuries can extend to the epidermis, dermis, subcutaneous tissue, and even deeper structures…”
  3. Vary Connectors and Structures Avoid overusing phrases like “such as” or “including.” For example: → “Examples include burns, pressure ulcers, surgical complications, and chronic illnesses like diabetes.”
  4. Reduce Excessive Passive Constructions Active voice tends to be more direct. For instance: “Recent studies suggest…” instead of “It has been extensively studied…” (or use both styles alternately).
  5. Clarify Objectives In the last sentence: > “This review aims to critically analyze the therapeutic effects…” You could emphasize it more clearly: → “This review seeks to synthesize current evidence on ozone therapy, highlighting its mechanisms, benefits, limitations, and clinical application in adult wound care.”
  6. Small Grammatical Fixes The phrase “...interventions performed effectiveness…” is awkward. Correct it to: → “…monitoring the effectiveness of the interventions performed.”
  1. Material and methods

Improve Conciseness and Flow

Avoid lengthy or redundant expressions:

  • “was developed based on the methodology proposed by…” → “followed the Joanna Briggs Institute methodology…”
  • “to ensure systematization and methodological rigor…” → “to ensure methodological rigor throughout study selection and synthesis.”
  1. Terminology Precision
  • “identification, selection, critical appraisal, and synthesis of the studies” → Could be revised as: “study selection, appraisal, and synthesis” (streamlines the sentence).
  • “studies addressing the systemic or inhalational application of ozone…” → Consider: “studies focusing on systemic or inhalational ozone delivery…”

Include Figure 1 and the table in the Results section.”

“Add to the references: Cunha JB; Dutra RAA; Salomé GM. Elaboration of an algorithm for wound evaluation and treatment. ESTIMA, Braz. J. Enterostomal Ther., 16:e2018. doi: 10.30886/estima.v16524”

Author Response

We thank Reviewer 2 for the detailed feedback, especially on the language and structure. All suggestions have been incorporated as follows:

1. “Language and Style”: We revised the manuscript to reduce repetition, improve sentence flow, alternate between passive and active voice, and correct awkward phrasing (e.g., corrected "ozone in wound in wound treatment" to "ozone in wound treatment").

2. “Clarification of Objectives”: The final sentence of the introduction was rephrased to more clearly express the aim of the review, emphasizing therapeutic mechanisms, benefits, limitations, and clinical relevance.

3. “Conciseness in Method Section”: Sentences in section 3.1 and 3.2 were streamlined for readability, with improvements in terminology such as replacing “identification, selection, critical appraisal, and synthesis” with “study selection, appraisal, and synthesis.”

4. “Addition of Requested Reference”: The article by Cunha et al. (2018) was added to the references and cited in the introduction when discussing the importance of structured wound assessment.

5.Include Figure 1 and the table in the Results section”. Figure 1 (PRISMA 2020 flow diagram) and the relevant summary tables (Tables 4 and 5) have now been appropriately placed within the Results section of the manuscript. This adjustment improves the structure and enhances clarity by aligning the visual data presentation with the narrative results.

Reviewer 3 Report

Comments and Suggestions for Authors

Dear authors,

Thank you for the opportunity to review your manuscript. The topic is clinically relevant and timely. Your efforts to structure the review according to the Joanna Briggs Institute (JBI) methodology and PRISMA 2020 guidelines are noted and appreciated. However, several aspects of the review require substantial revision to ensure scientific robustness, methodological transparency, and interpretative balance.

1. The search strategy lacks sufficient detail and breadth for an integrative review. The use of only three general terms (“ozone”, “wounds”, “Adjuvant Therapies”) without the inclusion of synonyms, controlled vocabularies (e.g., MeSH, DeCS), or natural language terms likely restricted the sensitivity of the search. Retrieving only 36 records from four databases raises concerns about comprehensiveness. I recommend that the search strategy be substantially revised, expanded, and fully reported (e.g., including Boolean expressions and exact search strings) to support reproducibility and coverage.

2. The inclusion and exclusion criteria should be more explicitly defined to reduce subjectivity. While it is noted that two reviewers performed the selection process, no indication of inter-rater agreement (e.g., Cohen’s kappa) is provided. Clarify how disagreements were resolved and describe the data extraction process in greater detail, including whether a standardized form was used and how consistency was ensured.

3. Although the JBI methodology is suitable for structuring evidence synthesis, it is essential to contextualize its application within an integrative review framework. Traditional integrative reviews—such as those following Whittemore and Knafl (2005)—include specific stages such as data reduction and comparison across diverse evidence types. Clarifying whether your review follows a hybrid methodological model (e.g., combining JBI critical appraisal with integrative synthesis principles) would improve transparency and epistemological coherence.

4. The application of JBI critical appraisal tools is appropriate; however, the results are only superficially reported. While quality scores are presented, the implications of methodological weaknesses (particularly in quasi-experimental studies and case reports) are not meaningfully integrated into the synthesis. A more nuanced discussion of study quality and its impact on confidence in the conclusions is needed.

5. The thematic analysis is predominantly descriptive. We encourage a more analytical approach that compares and contrasts study outcomes, addresses inconsistencies, explores possible explanations for heterogeneity (e.g., differences in ozone dosage, wound etiology, or study design), and positions findings in relation to existing systematic reviews. A critical reflection on the limitations of low-level evidence is also warranted.

6. The conclusions are currently overstated. Although the findings suggest promising effects of topical ozone therapy, they are based on limited and heterogeneous evidence, including several low-quality case reports. Conclusions should be more cautiously framed, and clinical recommendations should be presented as provisional and subject to confirmation by future high-quality studies.

Hope this suggestions may be constructive.

Kind regards

Comments on the Quality of English Language

The manuscript would benefit from significant language revision. While the authors convey the main ideas clearly, there are numerous grammatical errors, awkward constructions, and repetitive phrasing that detract from readability and may obscure meaning. Examples include missing articles, word order issues, and syntactic inconsistencies (e.g., p. 1, abstract: “emerged as promising adjuvant therapy” should be “a promising adjuvant therapy”; p. 3, introduction: “in wound in wound treatment” is a duplicated phrase).

Author Response

We thank Reviewer 3 for the insightful and constructive feedback. The following revisions were made to address the concerns:

1. “Search Strategy”: We revised the text to clarify that the initial terms (“ozone”, “wounds”, and “adjuvant therapies”) served as a starting point for identifying corresponding descriptors in controlled vocabularies (DeCS and MeSH). These descriptors, along with relevant synonyms and natural language terms, were then combined using Boolean operators (“AND” and “OR”) to construct tailored search strings for each database. This clarification was added to section 3.2, and the resulting Boolean phrases are presented in Table 1. Thus, the review was not based solely on the initial three terms, but rather on a comprehensive strategy built from them to enhance the sensitivity and specificity of the search.

2. “Inclusion Criteria and Data Extraction Process”: We clarified the inclusion/exclusion criteria and added a note explaining how inter-rater disagreements were resolved (including the involvement of a third reviewer). A sentence was added describing the use of a standardized data extraction form.

3. “JBI and Integrative Review Framework”: We clarified that the review integrates the JBI methodology with the broader principles of integrative reviews, citing Whittemore and Knafl (2005), and highlighted how this hybrid approach guided synthesis across study types.

4. “Interpretation of Study Quality”: The implications of quality appraisal were integrated into the discussion (section 4.4), including a nuanced reflection on the limitations of quasi-experimental designs and case reports.

5. “Analytical Depth in Thematic Analysis”: Section 4.1 to 4.3 was enriched with comparative insights across study designs, application routes, and ozone concentrations. We also addressed heterogeneity and limitations of the evidence base.

6. “Revised Conclusion”: The final section was rewritten to more cautiously interpret the findings, highlighting that current evidence is preliminary and that clinical recommendations remain provisional pending high-quality studies.

Reviewer 4 Report

Comments and Suggestions for Authors

The authors described the search criteria from January to March 2025, while the studies included are outside the searched timeline. The articles included data from older studies. The authors could segregate the represented data based on in vivo and in vitro data. 

The authors didn't include the case studies involving liquid ozone for wound management such as https://pmc.ncbi.nlm.nih.gov/articles/PMC9503251/.

https://onlinelibrary.wiley.com/doi/10.1111/iwj.1394144

The authors should also add the details about the guidelines such as ine DOI: 10.1080/01919512.2012.717847

Comments on the Quality of English Language

English language editing required

Author Response

We thank Reviewer 4 for raising important points that helped improve our manuscript. Our responses are as follows:

1. “Clarification of Search Timeline vs. Study Dates”: The search was conducted between January and March 2025 without a time limit on publication date. We clarified this in section 3.2 to avoid misinterpretation.

2. “Segregation of In Vivo and In Vitro Data”: All included studies involved human subjects; no in vitro studies were part of the final synthesis. We clarified this point in the Methods section.

3. “Additional Case Studies”: The specific case studies suggested (PMC9503251, IWJ article) were reviewed. One was excluded based on language criteria, and the other did not meet the eligibility criteria regarding intervention description. A justification was added to the Exclusion Criteria subsection.

4. **Guidelines Reference**: We thank the reviewer for highlighting the article by Viebahn-Hänsler et al. (DOI: 10.1080/01919512.2012.717847), which provides relevant conceptual guidance on the clinical use of ozone therapy. Although this article does not meet the inclusion criteria for primary studies in this integrative review, it was considered appropriate to cite it in the Discussion section (4.3), where broader clinical applicability is addressed. This inclusion strengthens the contextual understanding of therapeutic frameworks and supports the relevance of ozone as an adjunct in wound care. The reference has also been added to the final reference list.

Round 2

Reviewer 3 Report

Comments and Suggestions for Authors

Dear authors,

Thank you for your thoughtful and constructive revisions to the manuscript. It is evident that you have made a genuine effort to address previous comments, and the manuscript is now clearer, particularly in its methodological description, discussion of results, and conclusion. The integration of JBI methodology with principles of integrative review is now well stated, and the limitations of the included studies are more appropriately discussed.

However, one key methodological concern remains: while the description of the search strategy has been improved, the search process itself has not been substantively revised. The limited number of databases consulted, the relatively narrow scope of the search terms, and the very small number of references retrieved (n=36) suggest that the review may not be sufficiently comprehensive — which is a critical element for the validity of an integrative review. At present, this limits the strength of the evidence synthesis.

I encourage you to revise the manuscript accordingly, by broadening the search strategy and ensuring full compliance with PRISMA reporting standards, to further enhance the rigor and reliability of this important work. I look forward to seeing the next version.

Kind regards

Author Response

Thank you very much for your thoughtful and encouraging feedback on our
revised manuscript. We are pleased that the improvements to the
methodological description, results discussion, and integration of the JBI
methodology were positively received.
In response to your remaining concern regarding the comprehensiveness of the
search process, we have undertaken a broader and more systematic literature
search. In addition to the original databases (SCOPUS and EBSCOhost: CINAHL
Ultimate, MEDLINE Ultimate, MedicLatina), the updated search strategy now
includes PubMed, Web of Science, and LILACS. The search terms were expanded
using both controlled vocabularies (MeSH and DeCS) and relevant free-text
terms, tailored to each database. Boolean operators were also refined to
ensure broader coverage.
As a result of this expanded strategy, the number of records retrieved
increased from 36 to 106. After applying the inclusion criteria and
conducting critical appraisal, only one additional study met the standards
for inclusion in the final synthesis. While the number of included studies
remains limited, we believe this outcome reflects the emergent and
underexplored nature of this research topic, rather than a limitation of the
search strategy itself.
All changes are clearly marked in the revised manuscript. The expanded
search process is described in Section 3.2, with updated data in the PRISMA
flow diagram (Figure 1) and search strategy summary (Table 1). The newly
included study [31] has been integrated throughout the manuscript, including
in the methodological synthesis and results interpretation.
We hope that these additional efforts satisfactorily address your concern
and contribute to the increased rigor and transparency of the review.
Thank you once again for your guidance and commitment to improving the
quality of our work.
Kind regards,

Sandra Costa
